Molecular identification of avian influenza virus subtypes H5N1 and H9N2 in birds from farms and live bird markets and in respiratory patients

Tolba Hala M.N. 1
Abou Elez Rasha M.M. 2
Elsohaby Ibrahim 3 4
Ahmed Heba A. heba_ahmed@zu.edu.eg 2
1 Department of Avian and Rabbit Medicine, Faculty of Veterinary Medicine, Zagazig University , Zagazig , Egypt
2 Department of Zoonoses, Faculty of Veterinary Medicine, Zagazig University , Zagazig , Egypt
3 Department of Animal Medicine, Division of Infectious Diseases, Faculty of Veterinary Medicine, Zagazig University , Zagazig , Egypt
4 Department of Health Management, Atlantic Veterinary College, University of Prince Edward Island , Charlottetown , Prince Edward Island , Canada
Jiao Peirong
Electronic publication date: 2018 Sep 5
Publication date: 2018
Volume: 6
Electronic Location ID: e5473
Received 2018 Feb 27; Accepted 2018 Jul 28
Copyright: ©2018 Tolba et al.
Copyright year: 2018
Copyright holder: Tolba et al.
License: This is an open access article distributed under the terms of the Creative Commons Attribution License, which permits unrestricted use, distribution, reproduction and adaptation in any medium and for any purpose provided that it is properly attributed. For attribution, the original author(s), title, publication source (PeerJ) and either DOI or URL of the article must be cited.
License URL: https://creativecommons.org/licenses/by/4.0/

Keywords: Avian Influenza Viruses, H5N1, H9N2, Mutation

Funding: The authors received no funding for this work.

==============================
Background

Avian influenza viruses (AIVs) have been endemic in Egypt since 2006, and the co-circulation of high-pathogenic avian influenza H5N1 and low-pathogenic avian influenza H9N2 subtypes in poultry has been reported; therefore, Egypt is considered a hotspot for the generation of new subtypes and genotypes. We aimed to characterize AIVs circulating on commercial farms and in live bird markets (LBMs) during the winters of 2015 and 2016 in the study area and to identify H5N1 and H9N2 viruses in respiratory patients.

Methods

In total, 159 samples were collected from ducks, pigeons and quails on farms (n = 59) and in LBMs (n = 100) and screened by real-time RT-PCR for H5N1 and H9N2 subtypes. Clinical and postmortem examination was carried out on birds from the farms. Positive H5N1 samples were sequenced and analysed for mutations. Tracheal swabs were also collected from 89 respiratory patients admitted to respiratory hospitals in the same study area.

Results

Overall, H5N1 was identified in 13.6% of birds from farms, while it was detected in 17% of birds in LBMs. Subtype H9N2 was only identified from pigeons on farms (6.5%) and LBMs (11.4%). Sequencing of the haemagglutination gene (HA) in nine representative H5N1 isolates revealed a multi-basic amino acid motif at the cleavage site (321-PQGEKRRKKR/GLF-333), which is characteristic of highly pathogenic AIV, in five of our isolates, while the other four isolates showed an amino acid substitution (Q322K) at this cleavage site to make it (321-P K GEKRRKKR/GLF-333). All the isolates belonged to clade 2.2.1.2, and a comparison of HA sequences at the amino acid level showed 98.8–100% homology among the nine isolates, while they showed 94.1–96.1% identity with reference strains and the commonly used vaccine strain in Egypt. Out of 89 respiratory patients, 3.4% were positive for H5N1 and no patients were positive for H9N2.

Discussion

Our results indicated the circulation of the endemic H5N1 and H9N2 viruses among poultry in 2015 and 2016. Birds on farms and in LBMs are reservoirs playing a role in the dissemination of the virus and producing a public health risk. The application of proper hygienic measures in farms and LBMs to control the exposure of birds and humans to the source of infection along with continuous monitoring of the circulating viruses will provide information on understanding the evolution of the viruses for vaccine studies.

Introduction

Influenza A viruses belong to the family Orthomyxoviridae and are categorized into 16 haemagglutinin (HA) subtypes and nine neuraminidase (NA) types in wild and domestic birds (Fouchier et al., 2005). To date, genetic material of influenza viruses of the H17N10 and H18N11 subtypes has been detected in bats (Tong et al., 2012; Tong et al., 2013). In poultry, avian influenza viruses (AIVs) are classified according to the severity of the caused disease as highly pathogenic (HP) and low pathogenic (LP) viruses (Swayne & Suarez, 2000). Infection with LPAIVs is usually benign, including mild respiratory signs with/without a reduction in egg production. However, HPAIVs (especially H5 and H7 subtypes) may cause 100% mortality (Alexander, 2007).

The poultry industry in Egypt consists of two major sectors: commercial (estimated 850 million birds) and backyard (estimated 250 million birds) (Abdelwhab et al., 2010). As a result of insufficient poultry slaughterhouses and the preference for consuming freshly slaughtered birds, the poultry meat trade depends mainly on live bird markets (LBMs) selling live or slaughtered birds for consumers (Abdelwhab et al., 2010). There are two types of LBMs: small retail outlets that depend mainly on commercial poultry and, to a lesser extent, those that sell home-reared birds. The second type of LBM is a large market in each city and village that is held once per week and includes different vendors for selling birds, vegetables, seeds and other food. Birds of different species (chicken, ducks, geese, quail, pigeons) are usually put together in the same cage (Abdelwhab et al., 2010). The biosecurity standards and veterinary inspections in LBMs are minimal or completely absent. Birds in LBMs originate from different sources, such as farms and home rearing at various localities; which resulted in amplification and dissemination of the virus and increasing the probability of transmission to other birds and humans (Cardona, Yee & Carpenter, 2009). LBMs are considered critical links for commercial farms, slaughter houses, traders and final consumers (Ibrahim et al., 2007).

Since the H5N1 outbreak in Egypt in mid-February 2006, enormous loss in the poultry industry has occurred, and the slaughter campaign overwhelmed the resources of veterinary and public health authorities (Abdelwhab & Hafez, 2011). Despite vaccination and biosecurity measures have been implemented, the disease is still endemic in Egypt and affecting the poultry and public health sectors. Multiple clades of Egyptian A/H5N1 have been reported with at least two distinct genotypes: the 2.2.1.1 clade isolated mainly from vaccinated commercial poultry (and rarely from backyard birds) and the 2.2.1/C virus isolated from backyard birds, small-scale commercial poultry and humans (Abdelwhab et al., 2012; Arafa et al., 2012a). Clade 2.2.1.2 has predominated in Egypt since 2011, and the virus has mutations associated with increased binding affinity to human receptors, thus posing a public health risk (Arafa et al., 2016a).

In addition to the economically disastrous effect of AIV on the poultry industry, public health concerns arose due to the zoonotic potential of HPAI H5N1 (Naguib et al., 2015). Worldwide, during the 2006 outbreak of HPAI H5N1, 202 persons were infected, and 113 died; most of these victims were in direct contact with infected or dead birds (Wang et al., 2006; WHO, 2006). In Egypt, since 2003 till 2017, 359 cases and 120 deaths were reported due to H5N1 (WHO, 2017). Most studies relied on seroprevalence to determine which human cases had the H5N1 subtype (Gomaa et al., 2015). Poultry infected with the H9N2 AIV are usually asymptomatic or show mild clinical signs; however, frequent outbreaks of the virus have been reported in Asia and the Middle East with high mortality (Guo et al., 2000; Naeem et al., 1999), Morocco (El Houadfi et al., 2016) and Burkina Faso (Zecchin et al., 2017). The co-circulation of HPAI H5N1 and LPAI H9N2 has added an additional risk factor to the poultry industry and humans (Arafa et al., 2012c). Reassortment of these LPAI viruses with HPAI H5N1 could produce novel influenza viruses capable of crossing the interspecies barrier and causing zoonotic transmission and infection in humans (Turner et al., 2017). The isolation of H9N2 virus from humans in contact with poultry has been reported indicating its zoonotic potential (Kalthoff, Globig & Beer, 2010; Turner et al., 2017). In Egypt, since 2010, H9N2 has become endemic; furthermore, in 2015, the first human H9N2 case was reported (Naguib et al., 2015). Therefore, this study aimed to characterize AIVs circulating on commercial farms and in LBMs during 2015–2016 in the study area and to identify the H5N1 and H9N2 viruses in respiratory patients.

Materials and Methods

Sampling and data collection

Duck, pigeon and quail samples were collected during the winters of 2015 and 2016 from farms and LBMs in Sharkia Governorate, Egypt. The birds in LBMs were housed in the same cages. The birds from farms were those admitted to the Clinic of Avian and Rabbit Medicine Department, Faculty of Veterinary Medicine, Zagazig University, Egypt. Birds admitted to the clinic were subjected to clinical and postmortem examination. The number of sampled birds from LBMs and farms, respectively, included ducks (45 and 24), pigeons (35 and 31) and quails (20 and four). None of the birds from farms or LBMs received AIV vaccines except one duck sample from a farm (collected in 2015). Tracheal swabs from birds in LBMs were pooled (each pool consisted of swabs from five birds) and collected in tubes containing 2–3 mL viral transporting medium (VTM), which consisted of phosphate-buffered saline (PBS) (Biowhittaker) and PEN-STREP (10,000 U penicillin/ml—10000 U streptomycin/ml, Biowhittaker, lonza). From each bird admitted to the clinic, the brain, liver and lungs were pooled and homogenized in 10 mL sterile PBS. The homogenate was then centrifuged at 2,000 rpm for 20 min at 4 °C. The supernatant was pipetted and transferred to sterile Eppendorf tubes containing 100 µL PEN-STREP antibiotic and stored at −20 °C until used for real-time RT-PCR and virus isolation.

Throat swabs from 89 respiratory patients admitted to respiratory clinics in general hospitals in Sharkia Governorate, Egypt, were collected in VTM tubes. In addition to the sample collection, an epidemiological questionnaire was conducted to collect information about patients, including sociodemographic information (sex, age, occupation, residence, and education level), level of knowledge about AI, attitudes and practices. Informed verbal/written consent for participation in the study was obtained from all the participants, and the study was approved by the Committee of Animal Welfare and Research Ethics (protocol #12/2015), Faculty of Veterinary Medicine, Zagazig University, Egypt.

Virus isolation

Swabs and tissue homogenates from the examined birds were injected into 9–11-day-old specific pathogen-free embryonated chicken eggs via the allantoic route according to recommendations of the OIE (2012). The allantoic fluid was collected from eggs with dead embryos after 48 h and screened by slide haemagglutination test. At least three successive embryo passages were applied for each sample to be negative.

Real time RT-PCR

Viral RNA from HA allantoic fluids (bird samples) and from human swabs was extracted using the QIAamp viral RNA Mini kit (Qiagen, Hilden, Germany, GmbH) following the manufacturer’s guidelines. Primers and probes supplied from Metabion (Planegg, Germany) targeting the H5N1 (Aguero et al., 2007; Londt et al., 2008) and H9N2 (Ben Shabat et al., 2010) subtypes were used (Table S1). Real-time RT-PCR amplifications were performed in a final volume of 25 µL containing 7 µL of RNA template, 12.5 µL of 2x QuantiTect Probe RT-PCR Master Mix (Qiagen, Hilden, Germany), 3.625 µL PCR-grade water, 0.25 µL of each primer (50 pmol), 0.125 µL of each probe (30 pmol) and 0.25 µL of QuantiTect RT Mix. Reverse transcription was performed at 50 °C for 30 min, followed by primary denaturation at 94 °C for 15 min, followed by 40 cycles of denaturation at 94 °C for 15 s, annealing at 54 °C for 30 s and extension at 72 °C for 10 s. The reaction was performed in a STRATAGENE MX3005P real-time RT-PCR machine. Samples that were positive by H9N2 subtype real-time RT-PCR were further confirmed using the same primers in a conventional RT-PCR reaction to assure that the resulting amplification had the correct product size. Extracted RNA was transcribed to cDNA using the RevertAid H Minus First Strand cDNA synthesis kit (Fermentas Inc., Waltham, MA, USA) following the manufacturer’s instructions.

Sequence and phylogenetic analysis

PCR amplification of the H5 gene segment spanning the cleavage site was carried out using the primers and reaction conditions previously described (Slomka et al., 2007). Nine H5N1 isolates from the positive bird samples were selected for sequencing. Amplicons of 311 bp were purified using a QIAquick PCR Product extraction kit (Qiagen Inc. Valencia CA) according to the manufacturer’s guidelines. Sequencing was then carried out using Bigdye Terminator V3.1 cycle sequencing kits (Perkin-Elmer, Foster City, CA, USA) in an Applied Biosystems 3130 genetic analyser (HITACHI, Tokyo, Japan). The HA subtypes were identified by nucleotide BLAST (http://www.ncbi.nlm.nih.gov/BLAST) and submitted to GenBank with accession numbers KX228223, KX228224, KX228225 and KX228226 (winter 2015 samples) and MG725890, MG725891, MG725892, MG725893 and MG725894 (winter 2016 samples). Alignment of the nucleotide sequences with other HA gene sequences available in GenBank was performed by the Clustal W method using the MegAlign module of DNAStar software (Lasergene version 7.2; DNASTAR, Madison, WI, USA). The phylogenetic tree was generated using the neighbour-joining method in MEGA version 7 (http://www.megasoftware.net). The tree topology was evaluated by 1,000 bootstrap analyses.

Statistical analysis

Statistical analysis was performed using STATA version 15 for Windows (Stata Corp., College Station, TX, USA). The prevalence of AIV infection in live birds and human patients was estimated from the ratio of positive to the total number of samples tested. Because of the low number of cases in respiratory patients, the unconditional associations between AIV infection and independent variables (age, gender, occupation, residence, educational level, knowledge about AIV, route of infection, transmission, preventive measures and vaccinations) were assessed by Fisher’s exact test. Further, correspondence analysis was used to analyse the relationship between the outcome and categorized independent variables (Greenacre, 2017).

Results

Clinical and postmortem examination

The farm birds examined in the study were obtained from cases admitted to the Clinic of Avian and Rabbit Medicine Department, Faculty of Veterinary Medicine, Zagazig University. The ducks showed general signs of illness, such as loss of appetite and decreased feed and water consumption, and specific signs, such as nervous signs (twisting of the neck, trembling and torticollis), respiratory signs (rhinitis, nasal and ocular discharge) and greenish watery diarrhoea. Pigeons showed nervous signs, including twisting of the neck (Fig. 1A), inability to fly, tremors, incoordination, circling in flight, flying backwards and torticollis. Respiratory signs in some cases and greenish diarrhoea were also observed. None of the quails showed clinical signs except loss of appetite and low egg production.

Figure 1 Clinical manifestations and PM lesions of birds suspected to be infected with AIVs.

(A) Nervous signs (twisting of neck) in 3 months old pigeon, (B) necrosis in pancreas of two months old Balady duck, (C) congestion with petechial hemorrhage in the brain of 1.5 months old Muscovy duck, (D) congestion in spleen of 1.5 months old Muscovy duck, (E) congestion in the brain of 3 months old pigeon with petechial hemorrhage in the inner aspect of the skull, and (F) congestion in the liver of 1 year old Balady pigeon. Photo credit: Hala Tolba.

Postmortem examination of the ducks revealed congestion in the brain with petechial haemorrhage, congestion in the liver and spleen with necrosis and haemorrhage in the pancreas (Figs. 1B, 1C and 1D). In pigeons, the postmortem lesions showed septicaemia (congested parenchymatous organs) in most organs with congestion in the brain (Figs. 1E and 1F). Quails showed congestion in parenchymatous organs, with congestion in the brain. In female quails, salpingitis and congestion in the uterus were observed.

Virus isolation and haemagglutination

Inoculation of embryonated chicken eggs with swabs and tissue homogenates from the bird samples revealed mortality and AIV lesions in embryos within 24–48 h in all pigeons and 19 of the duck samples of farm origin. Haemagglutinating viruses were detected in 13 (41.9%) pigeons and six (25%) duck samples. With regard to samples from LBMs, inoculated eggs showed lesions in 15 pigeons, 17 ducks and four quail pooled samples. Haemagglutination was observed in 13 (37.1%) pigeons, 14 (31.1%) ducks and one (5%) quail sample. The HA unit of infective AIV allantoic fluids ranged from 1/64 to 1/1024.

Molecular identification

Positive HA allantoic fluids from pigeon, duck and quail samples were subjected to real-time reverse transcription (RT)-PCR for the identification of H5N1 and H9N2 subtypes. The results revealed that the H5N1 subtype was identified in ducks (16.7%) and pigeons (12.9%) of the farm origin, while H9N2 subtype was only identified in 6.5% of pigeons (Table 1). All the samples were from non-vaccinated birds except one duck sample (collected in 2015).

Table 1 Proportion of positive samples for H5N1 and H9N2 subtypes determined by RT-PCR.

Birds	Farms	LBMs	
	Number examined	H5N1 subtype	H9N2 subtype	Number examined	H5N1 subtype	H9N2 subtype	
Ducks	24	4 (16.7%)	0 (0%)	45	9 (20%)	0 (0%)	
Pigeons	31	4 (12.9%)	2 (6.5%)	35	7 (20%)	4 (11.4%)	
Quails	4	0 (0%)	0 (0%)	20	1 (5%)	0 (0%)	
Total	59	8 (13.6%)	2 (3.4%)	100	17 (17%)	4 (4%)	

Samples from LBMs identified the H5N1 subtype in 20% of both ducks and pigeons, while only 5% of quails were positive by real-time RT-PCR (Table 1). Subtype H9N2 was identified in 11.4% of pigeons only. Overall, H5N1 subtype was identified in 13.6% of birds from farms, while it was detected in 17% of birds from LBMs, which was a nonsignificant difference (P = 0.73). The Ct values from all the examined samples are shown in Table S2.

Sequence and phylogenetic analysis

H5N1 isolates were further confirmed by sequencing and nucleotide BLAST analysis. Sequencing of the HA segment revealed a multi-basic amino acid motif at the cleavage site (321-PQGEKRRKKR/GLF-333), which is characteristic of HPAIV, in five of our isolates (Fig. 2). The other four isolates showed an amino acid substitution (Q322K) at this cleavage site to make (321-P K GEKRRKKR/GLF-333). In comparison with the reference (parent, classic and variant) strains in Egypt, all the isolates in the current study showed the R325K amino acid substitution, while the Q322K substitution was found in only four of our isolates compared to the reference and vaccine strains.

Figure 2 Deduced amino acid sequences of the HA protein of our isolates (isolates from farms are marked by an asterisk) in comparison to parent (A/chicken/Egypt/06207-NLQP/2006), classic (A/chicken/Egypt/NLQP-0918/2009), variant (A/chicken/Egypt/0879/2008), vaccinal H5N1 (A/chicken/Egypt/Q1995D/2010), H5N2 (A/duck/Potsdam/1402-6/1986) strains and latest duck in Sharkia (A/duck/Egypt/1435CAS/2014), latest duck in Egypt (A/duck/Egypt/2/2015), latest pigeon in Sharkia (A/pigeon/Egypt/Sharkia-22/2014) and latest quail strains in Egypt (A/quail/Egypt/14102TCP/2014 and A/quail/Egypt/1171SG/2011).

Dots denote identical amino acids. The H5 influenza numbering was based on the alignment with A/Goose/Guangdong/1/96 (H5N1) minus the 16 amino acids known as HA signal peptide.

All the isolates belonged to clade 2.2.1.2 (Fig. 3), and a comparison of HA sequences at the amino acid level showed 98.8–100% homology among the nine isolates, while they showed 94.1–96.1% identity with reference strains and the commonly used vaccine strain (A/chicken/Egypt/Q1995D/2010) in Egypt. A lower similarity rate (90.2-91.6%) at the amino acid level was detected when compared with the A/duck/Potsdam/1402-6/1986 vaccine strain.

Figure 3 Phylogenetic analysis of HA gene nucleotide sequences of avian influenza virus (AIV) isolated from ducks, pigeons and quail, Sharkia, Egypt and other sequences available in GenBank.

The strains isolated in winter 2015 are marked by solid circles, while those of winter 2016 are marked with solid squares and vaccinal strains are marked with solid triangles. The tree was constructed via multiple alignments of 300-bp nucleotide sequence of HA gene using the neighbor-joining method and the Kimura-2-parameter model in MEGA7. The tree topology was evaluated by 1,000 bootstrap analyses.

Duck isolates in the current study showed 98.8–100% amino acid homology with the latest circulating duck strains in Sharkia Governorate (A/duck/Egypt/1435CAS/2014) and in Egypt (A/duck/Egypt/2/2015). In contrast, pigeon and quail isolates showed 97.7–100% amino acid identity with the latest pigeon circulating strain in Sharkia Governorate (A/pigeon/Egypt/Sharkia-22/2014) and 100% with the only two quail circulating strains in Egypt (A/quail/Egypt/1171SG/2011) and (A/quail/Egypt/14102TCP/2014).

Human patients

A total of 89 respiratory patients (age range 10–74 years), with a higher proportion of females (62.9%) than males (37.1%), were involved in this study (Table 2). Approximately 84.3% of participants had heard about AIV, 56.2% were rural residents, and only 10.1% of participants applied AIV preventive measures. Examination of throat swabs from respiratory patients by real-time RT-PCR revealed that three out of 89 (3.4%) of the patients were positive for the H5N1 subtype, while none of the samples were positive for H9N2. Figure 4 illustrates the risk factors associated with AIV infection among the respiratory patients admitted to hospitals. Positive cases were closely associated with illiterate older persons (≥50 years) who live in rural areas. The graph also highlights the close association between AIV cases and hygienic conditions (patients who did not apply any AIV preventive measures). On the other hand, patients without AIV infection (negative cases) were associated with young (≤19 years), educated, and hygienic persons who lived in urban areas.

Figure 4 Multiple correspondence analysis of risk factors associated with the avian influenza virus (AIV) infections in respiratory patients admitted to the general hospitals in Sharkia Governates.

Table 2 Demographic, knowledge, attitudes and practice data of 89 respiratory patients screened for H5N1 and H9N2 virus infection with real-time RT-PCR.

Variables	Collected samples, no. (%)	Influenza A-positive samples, no. (%)	P-value	
I. Demographic characteristics	
Gender				
Male	33 (37.1%)	0 (0%)	0.292	
Female	56 (62.9%)	3 (5.4%)		
Age				
≤19 years	25 (28.1%)	0 (0%)	0.253	
20–49 years	25 (28.1%)	0 (0%)		
≥50 years	39 (43.8%)	3 (7.7%)		
Educational level				
Illiterate	30 (33.7%)	3 (10%)	0.389	
Primary	9 (10.1%)	0 (0%)		
Secondary	17 (19.1%)	0 (0%)		
High school	15 (16.8%)	0 (0%)		
University	18 (20.2%)	0 (0%)		
Residence				
Rural	50 (56.2%)	3 (6%)	0.173	
Urban	39 (43.8%)	0 (0%)		
Occupations				
Employer	24 (27%)	0 (0%)	0.163	
Farmer	11 (12.4%)	0 (0%)		
Household	31 (34.8%)	3 (9.7%)		
Student	23 (25.8%)	0 (0%)		
II. Knowledge	
Had heard of AIV				
Yes	75 (84.3%)	3 (4%)	1.00	
No	14 (15.7%)	0 (0%)		
AIV is an infectious to human				
Yes	50 (56.2%)	1 (2%)	0.264	
No	7 (7.9%)	1 (14.3%)		
Don’t know	32 (35.9%)	1 (3.1%)		
Information source				
Television	58 (65.2%)	3 (5.2%)	1.00	
Internet	17 (19.1%)	0 (0%)		
None	14 (15.7%)	0 (0%)		
Knowledge about modes of AIV transmission	
Yes	46 (51.7%)	0 (0%)	0.109	
No	43 (48.3%)	3 (7%)		
Knowledge about AIV symptoms in human		
Yes	46 (51.7%)	0 (0%)	0.109	
No	43 (48.3%)	3 (7%)		
III. Attitudes and Practices	
Direct contact with to poultry				
Yes	56 (62.9%)	3 (5.4%)	0.244	
No	33 (37.1%)	0 (5.4%)		
Use of preventive measures during contact with poultry	
Yes	9 (10.1%)	0 (0%)	0.723	
No	80 (89.9%)	3 (3.8%)		
Healthcare-seeking behaviour				
Purchasing cold medicine by myself	62 (69.7%)	3 (4.8%)	0.550	
Seeking medical service from hospitals	27 (30.3%)	0 (0%)		
Influenza vaccine administration	
Yes	33 (37.1%)	0 (0%)	0.244	
No	56 (62.9%)	3 (5.4%)		
Total	89	3 (3.4%)		

Discussion

The endemic status of H5N1 and H9N2 viruses in Egypt has resulted in outbreaks in poultry and several human cases. The clinical examination of ducks naturally infected with H5N1 in our study revealed nervous signs and diarrhoea, while gross lesions, including congestion and haemorrhages, were observed in all internal organs. These were in accordance with different studies in Egypt (Hafez et al., 2010; Hagag et al., 2015). In Korea, the same observation in naturally infected ducks was reported (Rhyoo et al., 2015; Woo et al., 2011). The susceptibility of pigeons to infection with HPAI H5N1 has been controversial since 2004; however, other studies have documented that pigeons are experimentally susceptible to the virus and could serve as a source of infection for other animals (Mansour et al., 2014; Smietanka et al., 2011; Songserm et al., 2006). Clinical examination of pigeons in the present study revealed nervous signs, greenish diarrhoea and congestion in internal organs. This was in agreement with a recent study that reported the first natural infection of pigeons in Egypt (Mansour et al., 2014). The neurotropic nature of H5N1 in pigeons has been proven experimentally, supporting the observed neurological signs (Klopfleisch et al., 2006). Brain congestion was a predominant lesion in pigeons, and this finding has been reported previously in other studies (Jia et al., 2008; Mansour et al., 2014).

In the current study, the isolation rate of H5N1 and H9N2 viruses from birds originating from farms was insignificantly lower than the isolation rate from LBMs. Different studies have compared the isolation rates of AIVs from farms and LBMs; for instance, the detection rate of H5N1 in commercial farm birds (0.1%) was significantly lower than that (11.4%) of LBMs (El-Zoghby et al., 2013). In contrast, a higher detection rate in commercial farms (6.8%) versus 3.3% in LBMs was also reported in Egypt (Kayali et al., 2011). The high positivity rate of AIVs in LBMs in our study (17%) was comparable to 11.5% (El-Zoghby et al., 2013) and 12.4% (Abdelwhab et al., 2010) in Egypt, while lower rates of 0.08% in Nigeria (Joannis et al., 2008), 1.3% in Thailand (Amonsin et al., 2008) and 3.3% in Egypt (Kayali et al., 2011) were also reported. In poultry from commercial farms, AIVs were identified in 13.6% of the samples; in contrast, lower positivity rates were reported in other studies in Egypt (El-Zoghby et al., 2013; Kayali et al., 2011).

The detection of HPAI H5N1 in birds from LBMs has an epidemiological impact due to several factors: mixing birds of different origins, species and ages reduces the ability to trace the source of infection; inter- and intra-species transmission of the virus occurs; and importantly, there is increased exposure of vendors, consumers and children to sources of infection (Abdelwhab et al., 2010). Housing and mixing birds in cages for sale in LBMs plays role in the propagation and dissemination of AIVs (Turner et al., 2017).

All positive samples from LBMs were from birds with no symptoms, thus indicating the vital role of asymptomatic birds in the persistence and dissemination of the virus (Abdelwhab et al., 2010; Amonsin et al., 2008; Nguyen et al., 2005). Consumers prefer to purchase waterfowl, especially ducks, from LBMs and backyard rearing rather than from commercial farms. The higher detection rate of H5N1 virus from ducks in LBMs indicated the importance of this bird as a silent asymptomatic carrier that plays a role in dissemination and maintenance of the virus for long periods (El-Zoghby et al., 2013; ElMasry et al., 2017; Hassan et al., 2013).

Pigeons have been reported to be susceptible to infection with HPAI H5N1 and LPAI H9N2 viruses and could be a source of infection for other animals and subsequently for humans due to their abundance in backyard poultry farms (Mansour et al., 2014; Smietanka et al., 2011; Songserm et al., 2006). In our study, H5N1 and H9N2 viruses were detected in pigeons from both farms and LBMs. Studies reporting natural infection of pigeons are relatively scarce; however, experimental studies have reported the susceptibility of pigeons to HPAIV and their ability to disseminate the virus (Klopfleisch et al., 2006; Mansour et al., 2014; Smietanka et al., 2011). A few studies have reported the detection of H5N1 virus from naturally infected pigeons in Egypt during 2009 (Elgendy et al., 2016) and 2014 (Mansour et al., 2014; Mansour et al., 2017). The present study reported H5N1 infection in pigeons during 2015 and 2016 in Egypt. Subtype H9N2 was only identified in pigeons from farms (6.5%) and LBMs (11.4%); these birds were also co-infected with the H5N1 subtype. This was consistent with other studies in Egypt in different governorates (Kayali et al., 2014; Njabo et al., 2016).

In the present study, none of the quails from farms were positive for AIVs, while only one apparently healthy bird (5%) from a LBM was positive for H5N1. In accordance, a study conducted in Egypt from 2005–2007 reported the isolation of H5N1 from quails (Mady et al., 2010). Another study reported the confirmation of H5N1 from quails on farms (0.2%) and in LBMs (0.8%) (Arafa et al., 2016b). Quails have been reported to be an important mixing vessel for AIVs due to the expression of receptors for mammalian and avian AIVs on their epithelial cell surfaces (Nguyen et al., 2016; Thontiravong et al., 2012). The infection of quails with HPAI H5N1 has also been reported in different countries worldwide due to the intermingling of quails with other birds in LBMs (Nguyen et al., 2016). None of the quails were positive for H9N2, and this finding was also reported in a recent study (Zahida et al., 2017). In contrast to this finding, another study in Egypt described the isolation of H9N2 from apparently healthy quail (Arafa et al., 2012b; El-Zoghby et al., 2012).

The co-existence of H5N1 and H9N2 subtypes in poultry has been reported, which has caused public health concerns because the endemic H9N2 viruses in Egypt contain elements that may favour avian-to-human transmission (Kandeil et al., 2017; Sun et al., 2011). Moreover, the co-circulation of the two subtypes has resulted in reassortment between the two subtypes, and thus, the reassorted virus has been associated with increased mortality and spread of infection among poultry (Kayali et al., 2014). Therefore, Egypt is considered a hotspot for the generation of new subtypes and genotypes (Abdelwhab & Abdel-Moneim, 2015).

The isolates of the current study revealed the presence of a multi-basic amino acid sequence “EKRRKKR/GLF” at the proteolytic cleavage site, which is characteristic of HPAIV. This was consistent with other studies in the same area (Arafa et al., 2016a; ElBakrey et al., 2015; Mansour et al., 2014). This pattern has predominated since 2012, replacing the “ERRRKKR” pattern (Arafa et al., 2016a). All the isolates showed the R325K substitution at the cleavage site, which has been characteristic of 2.2.1.2 clade viruses since 2011 and is consistent with Arafa et al. (2016a). The diversity of the HA gene observed in the current study indicates the active circulation of the virus in the study area. This finding supports the genetic diversity of the H5N1 HA gene in Egypt (Arafa et al., 2016a).

Interestingly, Q322K substitution was obvious in four of our isolates compared to the reference and vaccine strains. This is indicative of continuous genetic evolution of the viruses. The higher similarity rate at the amino acid level among our isolates (98.8–100%) and higher identity between duck (98.8–100%), pigeon (97.7–100%) and quail (100%) when compared with the latest circulating isolates of each indicate the presence of a common ancestor progenitor. Similar observations were previously reported in the same study area (ElBakrey et al., 2015). Egypt has three major risk factors that contribute to the endemicity of the disease: a high density of domestic waterfowl, high density of the rural human population and abundance of water and irrigation resources (Abdelwhab & Hafez, 2011). The inadequate measures to control AIV in Egypt and the profound genetic drift in the HA gene have resulted in the endemic status of the disease (Kayali et al., 2016). Human cases of H5N1 were mostly caused by infection with viruses in subclade 2.2.1, which frequently circulate in LBMs and backyard poultry (Dudley, 2009).

Molecular detection of H5N1 and H9N2 viruses in 89 respiratory patients revealed that 3.4% were infected with only the H5N1 subtype. These findings are in agreement with a recent study in Egypt, where the H5N1 subtype was reported in 5.8% of respiratory patients (Hussien et al., 2017). The low detection rate of HPAI H5N1 in human patients was consistent with other findings, thus supporting the previously reported low rate of transmission from birds to humans (Njabo et al., 2016). However, other studies have reported that none of the human participants in contact with infected birds were positive for the virus or antigens (Ghoneim, Abdel-Moein & Zaher, 2014). They attributed this to the need for mutations in the haemagglutination protein in humans for successful transmission from birds (Maines et al., 2011).

The results of our study indicated that the three H5N1 cases were in contact with poultry in rural areas, which is a predisposing factor for acquiring H5N1 infection; similar findings were previously reported (Hussien et al., 2017). The disease occurs more frequently in rural areas in the Nile Delta due to backyard rearing of birds and more frequent contact with birds sold in LBMs (Njabo et al., 2016). No specific symptoms were observed in the patients with H5N1 infection, which is consistent with other studies (CDC, 2016; Hussien et al., 2017). Consequently, diagnosis of avian influenza infection cannot rely on clinical signs. The three positive cases in our study were adult females (>50 years). Kayali et al. (2011) reported that the case-fatality rate in humans infected with H5N1 was 34% and was significantly higher in female patients and increased with age. In contrast, the most affected demographic groups are young people and women (Njabo et al., 2016). Consistent with our results, recent H5N1 human cases reported exposure to poultry (70%), and most cases reported fever (98%), sore throat (94%) and cough (83%) (Kayali et al., 2016). The zoonotic potential of the H9N2 subtype and its co-existence with H5N1 have been documented in Egypt through occupational exposure (Gomaa et al., 2015). None of the human cases were positive for H9N2; however, a case was previously reported in South Egypt (OIE, 2015). Continuous monitoring and surveillance is essential to prevent this subtype from infecting humans (Njabo et al., 2016).

Conclusions

The findings of this study indicated the circulation of endemic H5N1 and H9N2 viruses among poultry in 2015 and 2016. Birds on farms and in LBMs are reservoirs playing a role in the dissemination of the virus and producing a public health risk. Proper hygienic measures should be applied on farms and in LBMs to control the exposure of birds and humans to the source of infection. Continues surveillance and monitoring of the circulating viruses is important for understanding the evolution of the virus and to better select viruses for vaccine studies to minimize the wide spread of the viral infection.

Supplemental Information

Data S1 Raw data

Click here for additional data file.

Supplemental Information 1 Human patient questionnaire—translated copy

Click here for additional data file.

Figure S1 Amplification curve of human samples for the identification of avian influenza viruses H5N1

Click here for additional data file.

Table S1 Sequences of the primers and probes used for avian influenza viruses

Click here for additional data file.

Table S2 Cycle threshold (Ct) values of the positive H5N1 and H9N2 positive samples from birds and humans

Click here for additional data file.

The authors are thankful to Dr. Shimaa M.G. Mansour, assistant professor of virology and vet. Ahmed Magdy, assistant lecturer of zoonoses, Faculty of Veterinary Medicine, Zagazig University for their valuable advice in analyzing the sequencing data.

Additional Information and Declarations

Competing Interests

Author Contributions

Human Ethics

DNA Deposition

Data Availability

The authors declare there are no competing interests.

Hala M.N. Tolba and Rasha M.M. Abou Elez performed the experiments, contributed reagents/materials/analysis tools, authored or reviewed drafts of the paper, approved the final draft.

Ibrahim Elsohaby analyzed the data, prepared figures and/or tables, authored or reviewed drafts of the paper, approved the final draft.

Heba A. Ahmed conceived and designed the experiments, analyzed the data, prepared figures and/or tables, authored or reviewed drafts of the paper, approved the final draft.

The following information was supplied relating to ethical approvals (i.e., approving body and any reference numbers):

The study was approved by the Committee of Animal Welfare and Research Ethics, Faculty of Veterinary Medicine, Zagazig University, Egypt (Protocol # 12/2015).

The following information was supplied regarding the deposition of DNA sequences:

GenBank with accession numbers KX228223, KX228224, KX228225 and KX228226 (winter 2015 samples) and MG725890, MG725891, MG725892, MG725893 and MG725894 (winter 2016 samples).

The following information was supplied regarding data availability:

The raw data are provided in the Supplemental Files.

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
