# Peer review of "Molecular identification of avian influenza virus subtypes H5N1 and H9N2 in birds from farms and live bird markets and in respiratory patients"

_PeerJ, doi:10.7717/peerj.5473_

## Round 0.1 · original submission · Major Revisions

We have received 4 reports on your submission. Please follow these comments to revise your manuscript, and then resubmit it.

In particular note the comments of Reviewer 4, and their appended PDF

Thank you!

Reviewer 1 ·

Basic reporting

English of the manuscript should be improved throughly.

Experimental design

The research is not met a high technical standard. Beside the cleavage sites, more molecular information of these viruses, such as genome sequences of eight segments, and their phylogenetic analysis should be provided to reveal the relationship among the strains.

Validity of the findings

The conclusions of the manuscript were not supported very solidly by the results.

Additional comments

The manuscript reported the molecular identification of AIVs in poultry from farms, LBMs and patients in Egypt. In general, little new knowledge has been gotten through the manuscript. Maybe it is appropriate to publish it in a local journal rather than in an international journal.

Special comments:
1.English of the manuscript should be improved throughly.
PCR -> RT-PCR
Line 44: identity -> identities
Line 57: types -> subtypes
Line 65: fresh -> freshly
Line 67-68;
LIne 74-75;
Line 80: Although -> Despite
Line 92: since 2003 till 2017
........
........
........

2.The positive rates of H5N1 and H9N2 in poultry from farms were 13.6% and 3.4%, respectively. Is there any relationship between positive viral results and the clinical and postmortem examination results of the birds? In other words, are pathologies of the birds which were showed in Fig. 1 caused by AIV infection?

3.Line 216-217: The birds from farms were admitted to clinic due to their diseases, and the birds in LBMs should be healthy. So, it might be meaningless to compare their positive rates of AIVs.

4.Line 220-227 and Fig 3: There were two kind cleavage sites in nine H5N1 strains. The virus origins (farms or LBMs) should be showed.

5.Phylogenetic analysis: Beside the cleavage sites, more molecular information of these viruses, such as genome sequences of eight segments, and their analysis should be provided.

6.Human patients: The viral genetic information of the H5N1 strains from patients should be obtained to establish the linkage between the virus strains from human and poultry.

Reviewer 2 ·

Basic reporting

The text is clear and well structured. English could be improved (i.e. line 74, line 90).
In Line 96 authors should cite the H9N2 subtype cases detected in Morocco and Burkina Faso adding the appropriate references.

Experimental design

The manuscript is not original but it is interesting because it describes the co-circulation of H5N1 and H9N2 subtype viruses in poultry in Egypt, in addition it reports the data collected from patients of a hospital.
Methods described are clear and detailed.
The phylogenetic analysis would have been more accurate if the authors had considered a longer fragment (amplicons of 311 bp are quite short) but it can be considered acceptable because the characterization of the viruses is not the first aim of the work.

Validity of the findings

The manuscript describes the situation in Egypt in 2015-2016, it refers to two-year-old data so it does not have a big impact but it can be useful for the comparison with more recent sequence data and for the understanding of the evolution of the avian influenza viruses in Egypt and in the African continent in general.
Results are discussed properly.
The authors describe accurately the findings in relation to the different species cosidered and the different environments analyzed (farm, LBM and hospital).

Reviewer 3 ·

Basic reporting

The manuscript was clearly reported, with sufficient field background and professional article structure. In addition, the manuscript is self-contained with relevant results.

Experimental design

The study aimed to investigate the dynamic evolution of H5N1 and H9N2 avian influenza viruses in Egypt. The experiments were well designed in this study. The study was performed with high technical & ethical standard, and the methods were described with sufficient detail.

Validity of the findings

The authors in this study determined the epidemic situation of H5N1 and H9N2 influenza viruses. Although the evolution of H5N1 and H9N2 influenza viruses has been reported by different researchers, this manuscript demonstrated that the circulation of H5N1 and H9N2 viruses are still undergoing in Egypt. Therefore, this study suggests that intensive surveillance of avian influenza viruses should still be strengthened in Egypt. The data is convincing, and the conclusion is well stated.

Additional comments

1. line 83, "genotypes", should it be "clades"?
2. lines 82-87, the authors mentioned that two clades of H5N1 viruses were present in Egypt. However, they mentioned three clades, 2.2.1.1, 2.2.1/C and 2.2.1.2. What is the relationship among these three clades?
3. line 153, change "primers" into "PCR amplification".
4. line 367, change "finding" into "findings".

Reviewer 4 ·

Basic reporting

In this study by Tolba et al., the authors have characterized the avian influenza viruses which are circulating in the commercial farms and the live bird markets during 2015-2016. The samples were taken from ducks, pigeon and quail that were admitted to the Clinic of Avian and Rabbit Medicine Department. The authors did a systematic screening of the samples by Real-time RT-PCR for high pathogenic avian influenza (HPAI) H5N1 and low pathogenic avian influenza (LPAI) H9N2 virus targets. The positive samples were subjected to virus isolation and molecular identification. They have identified the viruses as H5N1 and H9N2 on the basis of HA sequencing and phylogenetic analysis.

The strength of the study is the animal human interface aspect. The authors made an attempt to identify the virus subtypes from the respiratory patients. They have shown that 3.4% of patients were positive for H5N1 virus, but none for H9N2 virus.

In general, this is an important study considering that HPAI H5N1 and LPAI H9N2 viruses are endemic in Egypt. Importantly, HPAI H5N1 virus poses a threat to both animal husbandry and human health and with the circulation of the LPAI H9N2 virus there is a risk of generating recombinant viruses. The study is well planned and the manuscript is well written. However, it is not clear whether the authors made any attempt to isolate viruses from the human samples that are positive for H5N1 virus by Real-time RT-PCR. It is important to sequence and characterize these viruses and determine the phylogenetic relation with the viruses isolated from the birds. In addition, there are few points which need to be addressed and the manuscript needs to be revised on those lines.

Experimental design

Specific comments:
1) Abstract: Line 26- I suggest inclusion of ‘High-pathogenic avian influenza (HPAI)’ for H5N1 subtype and ‘Low-pathogenic avian influenza (LPAI)’ for H9N2 subtype.
2) Abstract: Line 45-‘Out of 89% respiratory patients, 3.4% were positive for H5N1’. I wonder what percentages of patients were positive for H9N2. This is missing in the abstract. Only after reading through the whole manuscript we know that there are none. A line will be informative.
3) Abstract: Lines 48-51- ‘We recommend the ….vaccine studies’. It is well known that application of proper hygienic measures helps in control of the AI virus. I feel that the authors are not adding any new change in the system. Thus ‘we recommend’ can be removed and the statement rephrased as ‘The application of proper hygienic measures in farms and LBMs to control the exposure of birds and humans to the source of infection along with continuous monitoring of the circulating viruses will provide information on understanding the evolution of the virus and selection of viruses for vaccine studies’.
4) It is not clear if the authors came across any reassortants of H5N1 and H9N2 viruses in the bird samples in Egypt. This may be added in the discussion.
5) Introduction: Lines 56-57; the author mentions about 16 HA and 9NA subtypes and I am surprised about the mention. Two new HA and NA subtypes have been isolated from bats recently and that has been added to the influenza HA and NA subtypes. (Reference# Proc Natl Acad Sci U S A. 2012 Mar 13;109(11):4269-74. doi: 10.1073/pnas.1116200109. Epub 2012 Feb 27; PLoS Pathog. 2013;9(10):e1003657. doi: 10.1371/journal.ppat.1003657. Epub 2013 Oct 10; Virology. 2015 May;479-480:234-46. doi: 10.1016/j.virol.2015.03.009. Epub 2015 Mar 24.) I suggest rephrasing the sentence with the updated information and references.
6) Materials and Methods: It is important to know whether the birds (pigeon, quails and ducks) were housed in the same cage or kept independently. This should be mentioned in the respective section as well as in the discussion.
7) Materials and Methods: Though the authors have quoted references for the primers, it will be useful for the readers to have a list of the primers used as a Supplementary table.
8) Materials and Methods: Line 156; Does conventional reaction mean conventional RT-PCR? Correct the text. Which kit was used for the conventional RT-PCR?
9) Results: Line 200- What was the HA titer of the viruses isolated and the HA unit for the 13 pigeons, 14 ducks and 1 quail samples. This could be presented as a Supplementary Table. It is also not clear whether the authors attempted to isolate virus from the human samples which they got positive for H5N1 target?
10) Results: Line 215- Table-1 only depicts the % positivity however what was the Ct value is not clear. In my opinion showing the profile of Real-time RT-PCR or mentioning the Ct values of the positive samples would be informative.
11) Results: Lines 230-238; I wonder whether the authors included the strain mentioned in the text in the phylogenetic analysis since the reviewer could not locate the strains (Potsdam/1402; Egypt/1435; Egypt/2; Egypt/Sharkia-22 and Egypt/1171SG) in the tree Fig .3.
12) Results: Line 240- In this section, as mentioned above, it is not clear whether the human samples were subjected to virus isolation (same point raised above in #9). What was the HA unit? What was the Ct value of the human samples that were positive for H5N1 virus? How can the authors be sure that it was not due to contamination from the handling of the H5N1 positive bird samples? It is important to show the Ct value and the Real-time RT-PCR profile of the human samples.
13) Results: Considering the important relevance of animal human interface of avian influenza viruses, why were the human isolates not included in the Phylogenetic tree for analysis? It is important to sequence and characterize these viruses and determine the phylogenetic relation with the viruses isolated from the birds.
14) Discussion: Well written, however too lengthy, can be reduced by a page and some points such as about the reassortants need to be discussed. The authors need to add if they came across any reassortants of H5N1 and H9N2 viruses in the bird samples in Egypt.
15) Discussion: Lines 308-311; ‘The present study was the only in Egypt…. H5N1 subtype’. It is too much of an extrapolation to comment from the present study. I suggest that the statement be rephrased. Regarding the next line about co-infection I wonder how the authors have confirmed the co-infection of H5N1 and H9N2 viruses. Have they isolated both the viruses from the same bird samples? Or was the appearance of H5N1 a mere speculation which could be due to contamination during RNA isolation or Real-time RT-PCR? The co-infection must be established. The Real-time RT-PCR profile of all the targets H5, N1, H9 and N2 may be shown together and secondly, this can be clarified by isolating both the viruses from the same sample and then determine the HA unit of the viruses.
16) Figure 3: The vaccinal strains can be color coded for easy identification in the tree. Many strains mentioned in the text are missing from the tree. No human samples have been included in the analysis.

Minor comments:
1) Line 74: delete ‘is’
2) Line 78: change losses to ‘loss’
3) Line 80: delete ‘the’
4) Line 100: add ‘virus’ after H9N2
5) Line 119: delete ‘lonza’
6) Line 125: delete ‘s’ from ‘samples’
7) Line 134: Add ‘the’ before ‘QIAamp…
8) Line 135: Change ‘manufacturer’ to ‘manufacturer’s’
9) Line 137: Add ‘Real-time RT’ before PCR…
10) Line 138: Add the company name for the Quantitect
11) Line 146: Change ‘real-time PCR’ to ‘Real-time RT-PCR’. The reviewer suggests that this should be changed at all places in the text.
12) Line 156: Delete ‘was’
13) Line 178: change ‘routs’ to ‘route’
14) Line 211: Add ‘the’ before farm origin
15) Line 212; Add ‘subtype’ after H9N2
16) Line 216: Add ‘subtype’ after H5N1
17) Line 217: Change ‘in’ to ‘birds from’
18) Line 220: Change ‘belonged’ to ‘belonging’
19) Line 230: Change ‘96.1-94.1%’ as ‘94.1-96.1%’
20) Line 269: Add ‘viruses’ after H9N2
21) Line 284: Change ‘plays’ to ‘play’ and delete ‘a’
22) Line 290: Add ‘virus’ after H5N1
23) Line 295: Add ‘virus’ after H5N1
24) Line 301: Add ‘virus’ after H5N1
25) Lines 303, 321, 323: Add ‘viruses’ after H9N2
26) Line 314: Add ‘virus’ after H5N1
27) Line 390: Change ‘continues’ to ‘continuous’
28) The title in Table 1, please add ‘H5N1’ and ‘H9N2’subtypes
29) In Table 2 line 2, add ‘virus’ after H9N2 and ‘Real-time’ before RT-PCR

Validity of the findings

Virus isolation from the human samples.
Confirmation of coinfection of H5N1 and H9Nr viruses in bird samples.

Annotated reviews are not available for download in order to protect the identity of reviewers who chose to remain anonymous.

---

## Round 0.2 · Minor Revisions

Please revise your manuscript according to the reviewer's advice. Thank you.

Reviewer 4 ·

Basic reporting

No comment

Experimental design

No comment

Validity of the findings

Validity of coinfection in birds due to H5N1 and H9N2 viruses not clear.

Additional comments

In this revised manuscript by Tolba et al., the authors have either addressed to or answered the questions raised by the reviewer. There is only one query this reviewer is not clear about.
1) The authors mentioned in response to query R4.16 that both H5N1 and H9N2 viruses were isolated from the same bird samples. Each PCR was run separately. The H5N1 viruses were isolated, while, H9N2 were not successfully isolated due to low virus quantity (High Ct values), but to confirm that the H9 subtype was present, conventional PCR was carried out to ensure that the product was amplified.

The reviewer has a point that if due to low virus quantity H9N2 virus could not be isolated, how conventional PCR was successful. The reviewer would like to see the amplification results of both H5N1 and H9N2 viruses from the same bird samples.

Annotated reviews are not available for download in order to protect the identity of reviewers who chose to remain anonymous.

---

## Round 0.3 · accepted · Accept

I confirm that your article is now Accepted. Congratulations.

#